# Probability of Starting Two-Drug Regimen (2DR) vs. Three-Drug Regimen (3DR) in ART-Naïve and ART-Experienced Person with HIV (PWH) Across the First Wave of COVID-19 Pandemic

**DOI:** 10.3390/v16121822

**Published:** 2024-11-23

**Authors:** Alessandra Vergori, Nicola Gianotti, Alessandro Tavelli, Camilla Tincati, Andrea Giacomelli, Elena Matteini, Giuseppe Lapadula, Lucia Taramasso, Loredana Sarmati, Antonella D’Arminio Monforte, Andrea Antinori, Alessandro Cozzi-Lepri, on behalf of the ICONA Foundation Study Group

**Affiliations:** 1Viral Immunodeficiency Unit, National Institute for Infectious Diseases L. Spallanzani, Istituto di Ricovero e Cura a Carattere Scientifico (IRCCS), 00143 Roma, Italy; andrea.antinori@inmi.it; 2Infectious Diseases, Istituto di Ricovero e Cura a Carattere Scientifico (IRCCS), San Raffaele Scientific Institute, 20133 Milano, Italy; gianotti.nicola@hsr.it; 3Icona Foundation, 20133 Milan, Italy; alessandro.tavelli@icona.org (A.T.); antonella.darminio@unimi.it (A.D.M.); 4National PhD Programme in One Health Approaches to Infectious Diseases and Life Science Research, Department of Public Health, Experimental and Forensic Medicine, University of Pavia, 27100 Pavia, Italy; 5Clinica di Malattie Infettive, Dipartimento di Scienze della Salute, ASST Santi Paolo e Carlo-Presidio Ospedaliero San Paolo, Università degli Studi di Milano, 20133 Milano, Italy; camilla.tincati@unimi.it; 6Department of Biomedical and Clinical Sciences, Università degli Studi di Milano, 20133 Milano, Italy; dott.giacomelli@gmail.com; 7III Infectious Disease Unit, ASST-Fatebenefratelli Sacco, 20133 Milano, Italy; 8Dipartimento di Sicurezza e Bioetica, Università Cattolica del Sacro Cuore, 00143 Roma, Italy; elena.matteini@outlook.it; 9IRCCS Fondazione San Gerardo, Monza, Università di Milano Bicocca, 20133 Milano, Italy; giuseppe.lapadula@unimib.it; 10Infectious Disease Clinic, Istituto di Ricovero e Cura a Carattere Scientifico (IRCCS), Policlinico San Martino Hospital, 16132 Genova, Italy; taramasso.lucia@gmail.com; 11Clinical Infectious Diseases, Department of System Medicine, Tor Vergata University, 00143 Roma, Italy; sarmati@med.uniroma2.it; 12Centre for Clinical Research, Epidemiology, Modelling and Evaluation (CREME), Institute for Global Health, Univesity College London, London WC1E 6BT, UK; a.cozzi-lepri@ucl.ac.uk

**Keywords:** HIV, COVID-19 lockdown, 2DR, 3DR

## Abstract

**Background**: This study examined the impact of the COVID-19 lockdown on antiretroviral therapy (ART) prescriptions among persons living with HIV (PWH) in Italy. **Methods**: Data from the ICONA cohort included ART-naïve individuals who started ART between January 2019 and December 2022, and ART-experienced individuals who started new ART with HIV RNA ≤50 cps/mL from January 2016 to December 2022. The analysis focused on the proportion of PWH starting or switching to dual (2DR) versus triple (3DR) ART regimens. Comparisons were made using Chi-square and Kruskal-Wallis tests, with logistic regression (LR) to assess associations, adjusting for sex and age. **Results**: Among 2481 ART-naïve PWH, 17% were female, with a median age of 40. Using 2020 as the comparator (the lockdown year), the odds ratio (OR) from fitting a LR showed a reduced probability of prescribing 2DR both before and after 2020. The proportion of PWH starting 2DR was 9% in 2019, 18% in 2020, 13% in 2021, and 10% in 2022. Among 12,335 ART-experienced PWH, 20% were female, with a median age of 47. The proportion switching to 2DR rose from 24% in 2016 to 38% in 2020, 62% in 2021, and 65% in 2022, showing a >3-fold higher probability to be switched to 2DR instead of 3DR in recent years (2021-2022). **Conclusions**: For ART-naive PWH, 2DR initiation did not decrease during the 2020 lockdown but changed in the following years, possibly indicating shifts in clinical practice or resuming HIV services. For ART-experienced PWH, 2DR prescriptions increased significantly over time, especially for INSTI-based regimens.

## 1. Introduction

Current guidelines for managing people living with HIV (PWH) emphasize timely care engagement, initiating antiretroviral therapy (ART) with either a three-drug regimen(3DR) or a dual therapy regimen (2DR), regular medical appointments, and high adherence to treatment [1,2] These measures are essential for preventing HIV-related morbidity, mortality, and transmission [3,4,5].

The COVID-19 pandemic in 2020 caused widespread healthcare disruptions, including the cancellation or postponement of routine appointments. PWH, who need regular healthcare engagement for viral suppression, were particularly affected [6,7,8].

During the first wave of the pandemic, the British HIV Association (BHIVA) recommended maintaining current HIV treatments in ART-experienced PWH and initiating bictegravir/emtricitabine/tenofovir alafenamide (B/F/TAF) for all ART-naïve PWH, unless contraindicated [9]. B/F/TAF was chosen for its high resistance barrier, minimal side effects, no need for kidney function monitoring, minimal drug interactions, and no food requirements [10,11,12]]

Given these interim recommendations and the absence of specific therapeutic guidelines in Italy during the pandemic, we hypothesized that service disruptions and the BHIVA guidance may have influenced Italian clinicians to favor 3DR over 2DR around 2020, and thus expect a reduction in dual therapy use during this period.

## 2. Materials and Methods

This retrospective observational multicenter study was conducted in 61 centers from the Italian Cohort Naïve Antiretroviral (ICONA) network [13], covering nine Italian regions.

### 2.1. Study Population

PWH aged 18 years or older seen for care at one of the ICONA Network participating sites: (a) ART-naïve with an ART initiation with 3DR or 2DR over the period of January 2019 to December 2022; (b) ART-experienced who switched to 3DR or 2DR with HIV-RNA < 50 cps/mL over the period of January 2016 to December 2022. HBsAg-positive individuals were excluded from the analysis. The temporal boundaries were chosen so that they spanned across the first pandemic wave and were based on the year of introduction of 2DRs for use in HIV treatment guidelines and the clinics.

### 2.2. Statistical Analysis

The exposure of interest was the year of starting/switching ART. This was defined as a categorical variable with groups 2019, 2020, and 2021–2022 for the ART-naïve population and 2016–2018, 2019–2020, and 2021–2022 for the ART-experienced participants. Chi-square test and Kruskal Wallis tests were used to compare the characteristics of participants according to the calendar period of start/switch for categorical and continuous factors, as appropriate. Logistic regression models were used to estimate the odds ratios (OR) of prescribing 2DR vs. 3DR according to the calendar period after adjusting for sex at birth age, nation of birth, level of education, employments status (plus line of therapy in the ART-experienced group). A sensitivity analysis including only ART-experienced patients switching to an INSTI-sparing regimen has also been performed by means of a separate logistic regression model. We also investigated whether the effect of calendar year might vary due to INSTI use, sex, and CD4 count at initiation by formally testing for interactions in the model. By restricting the analysis to INSTI-sparing regimens, we also controlled for the drug class of the anchor drug, as the majority of 2DRs were DTG-based.

### 2.3. Ethics

The ICONA study was approved by the Institutional Review Boards (IRBs) of all the participating centers. In accordance with the Italian legislation of observational studies, the last amendment of the ICONA Study was centrally approved by the IRB of the coordinating center INMI Lazzaro Spallanzani (CET Lazio Area 4, approval number 83-2024) and notified the Ethics Committee of each participating clinical center. All PWH signed a written consent form to participate in the study and for the processing of personal and clinical data, in accordance with the ethical standards of the committee on human experimentation and the Helsinki Declaration (last amendment October 2013).

## 3. Results

### 3.1. General Characteristics of the Study Population—ART-Naïve PWH

We included 2481 treatment-naïve PWH; 870 started ART in 2019, 522 in 2020, and 1089 in 2021–2022. Overall, 12% started with a 2DR (the majority with DTG/3TC) and 88% with a 3DR (mainly with TAF/F/BIC 43%, followed by TAF/F+DTG 15% and by TAF/F/DRV/c 9.3%) and 83% were males, with a median age of 40 years (interquartile range, IQR 32-51); their main modality of HIV acquisition was unprotected homosexual intercourse (47%). No evidence for a difference in the case mix of the participants across the three calendar period groups was observed, except for the proportion of participants with HIV RNA > 100,000 copies/mL at ART-start, which was higher in 2021–2022 than in previous years (24.5% versus 20% in 2020 and 2019; *p* = 0.02) (Appendix A).

### 3.2. Proportions of ART-Naïve PWH Starting a 2DR, According to Calendar Period

The proportions of participants starting a first-line ART with 2DR remained substantially stable throughout the period analyzed (Figure 1), settling around 10%, with little evidence for a change over time (Cochran–Armitage test *p* = 0.224). However, a peak of 2DR prescriptions was identified in 2020 (17.8%) (*p* < 0.001).

After controlling for the potential confounding by gender, age, nation of birth, level of education and employment, in a logistic regression model and using the year 2020 as the reference category, there was evidence that the proportion of participants starting a 2DR was lower both in 2019 (OR = 0.46; 95%CI 0.34, 0.64) and in 2021-2022 (OR = 0.63; 95%CI 0.47, 0.84), Appendix A).

### 3.3. General Characteristics of Study Population—ART-Experienced Virologically Suppressed PWH

We included 12,335 virologically suppressed ART-experienced PWH in this analysis: 7082 switched ART in 2016–2018, 3306 in 2019–2020, and 1947 in 2021–2022. Overall, 20% were females; participants had a median age of 47 years (38–55); 46% reported unprotected homosexual intercourse as their modality of HIV acquisition, followed by heterosexual contacts (39%). The majority, 74%, were simplified to a 3DR (mainly TAF/F/RPV 19.7%, followed by TAF/F/BIC 19.1%, by TAF/F/EVG/c 14%, and by ABC/3TC/DTG 11%) and the remaining 26% to a 2DR (mainly DTG/3TC 61%, followed by RPV/DTG 12.8%). At the time of switching ART, 3% had a CD4+ count of less than 200 cells/mL; the proportion of PWH with a baseline CD4+ count of less than 200 cells/mL was slightly higher in the calendar period 2016–2018 compared to other periods (3% vs. 2%; *p* = 0.001); other characteristics with relevant differences between the three calendar periods were mode of transmission, with a higher proportion of injecting drug users in the recent period (*p* < 0.001) and baseline eGFR with a higher proportion of PWH with eGFR < 60 mL/min/1.73 m^2^ in 2021–2022 (*p* < 0.001) (Appendix A).

### 3.4. Proportions of ART-Experienced Virologically Suppressed PWH Switching to a 2DR vs. 3DR According to Calendar Period

The proportion of virologically controlled ART-experienced participants who were switched to a 2DR significantly increased throughout the period (and conversely, 3DR significantly decreased) reaching more than 60% of a switch towards 2DR in 2021 and 2022 (Figure 2) (Cochrane-Armitage test *p* < 0.0001). After controlling for gender age, nation of birth, level of education, employment and line of therapy previously received in a logistic regression model, the trend in the odds of switching to 2DR vs. 3DR appeared to be completely reversed before and after the pandemic. Compared to 2019–2020, the OR of switching to 2DR vs. 3DR was 0.35 (95% CI: 0.32–0.39) in the pre-pandemic years 2016-2018, and 3.75 (95% CI: 3.33–4.22) in the post-pandemic years 2021–2022 (Appendix A).

After restricting the analysis to INSTI-spared regimens alone, after the initial peak of 22% in 2017, since 2017, switches to 2DR remained substantially stable at approximately 9% throughout the period (Cochrane-Armitage test *p* < 0.0001) (Appendix A). This subset analysis showed a higher prescription of 2DR non-INSTI in 2016-2018 (vs 2019–2020: OR 1.56; 95%CI 1.13, 2.16) and confirmed the trend in recent years [OR 0.98; 95%CI 0.59, 1.62)] (Appendix A).

## 4. Discussion

This study examined temporal trends in antiretroviral therapy (ART) prescription patterns, specifically dual therapy regimens (2DR) versus triple therapy regimens (3DR), among over 13,000 people living with HIV (PWH) routinely seen for care in Italy, before and after the initial wave of the COVID-19 pandemic and including both ART-naïve and ART-experienced PWH. Global HIV treatment guidelines recommend immediate initiation of ART upon diagnosis to prevent HIV-related morbidity, mortality, and transmission [1,2,4,14] offering a range of INSTI-based 2DRs or 3DRs based on efficacy and tolerability data from clinical trials [1,2]. During the COVID-19 pandemic, the British HIV Association (BHIVA) issued temporary pragmatic recommendations in May 2020, advising caution but not overriding national health policies. Despite these guidelines, our analysis found a higher prevalence of 2DR use among ART-naïve PWH in Italy in 2020, followed by a subsequent decrease in usage in the following years. The reasons for this trend remain unclear and may reflect changes in recommendations or increased caution in the post-pandemic period. In our ART-experienced virologically controlled population, we observed an increasing frequency of 2DR vs. 3DR initiation, especially in recent years. Therefore, these data are even less consistent with our hypothesis that the COVID-19 pandemic and/or temporary modification of clinical services might have reduced the propensity of Italian physicians toward prescribing 2DR over the lockdown period. Of note, our analysis also shows that INSTI-sparing 2DR was popular in 2016 (>20%) but its use has subsequently dropped to approximately 8% and remained stable; in contrast, the prescription of 2DR INSTI-based regimens, mainly DTG and 3TC, appeared not to be affected by structural changes in services during the pandemic and its use seemed to have increased over time because of the availability of robust long-term results on the effectiveness and tolerability of these regimens from both randomized clinical trials and observational studies [15,16,17,18,19]. In contrast, we found no evidence that the frequency of prescription of 2DRs vs. 3DRs over time varied by sex and CD4 count at ART line initiation. Our analysis has some limitations. First, the analysis is mainly descriptive, and only key confounders such as age, sex, nation of birth, level of education, employment status and previous history of ART use have been taken into account in the regression adjustments; furthermore, both residual and unmeasured confounding may have biased the observed temporal trends. For example, even in Italy, where access to care is universal, access to specific drug regimens may vary by socioeconomics and the pandemic had a larger impact among the socially deprived. Our analysis controlled for nation of birth, level of education and employment status but residual confounding cannot be ruled out. Second, the analysis focuses on the calendar year as the main exposure of interest, and although we also studied possible effect measure modification by use of INSTI, sex, and CD4 count, our data do not identify full profiles of PWH initiating 2DRs and whether these have changed over time. Lastly, the analysis was conducted using time-windows of 1 year in length, and therefore, we cannot rule out that minor modifications might have been detected using a finer classification of the exposure.

## 5. Conclusions

Contrary to our initial hypothesis of a decreased trend of 2DR in PWH over the first wave of the COVID-19 pandemic, our data show its stable use in ART-naïve and even an increased trend in the ART switch setting. Although pragmatic recommendations dictated by the emergency situation during the COVID-19 pandemic were commendable, they did not seem to have had a large impact, at least in Italy.

## Figures and Tables

**Figure 1 viruses-16-01822-f001:**
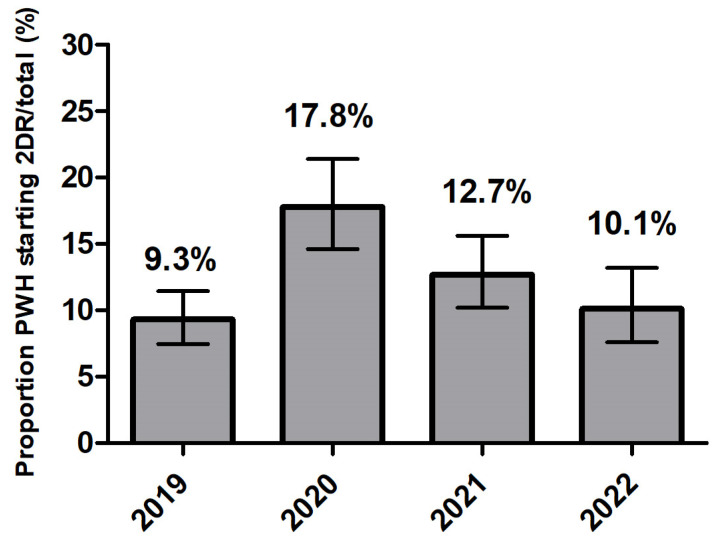
Proportion of PWH starting a two-drug regimen (2DR) 300/2481 (12%) according to calendar year in ART-naïve persons with HIV (proportion of 2DR over total number of regimens started).

**Figure 2 viruses-16-01822-f002:**
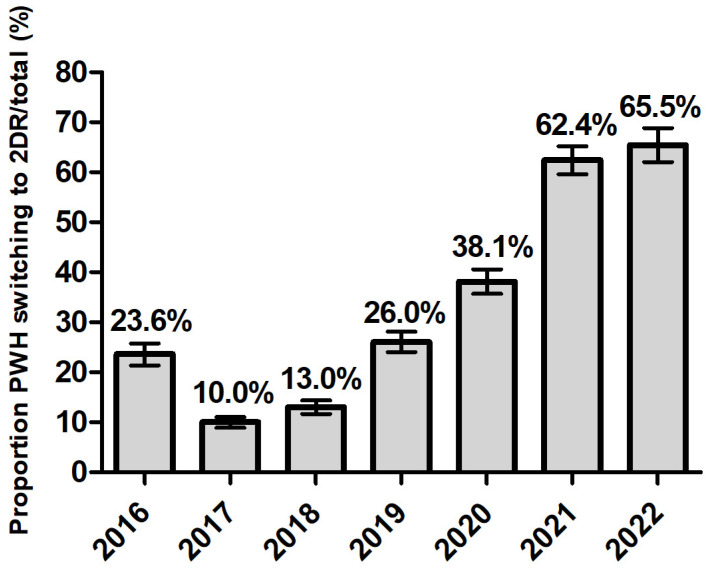
Proportion of PWH starting a two-drug regimen (2DR) 3259/12,335 (26%) according to calendar year in ART-experienced persons with HIV.

## Data Availability

Data will be available upon reasonable request to the corresponding author.

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
