# Peer review of "Probability of Starting Two-Drug Regimen (2DR) vs. Three-Drug Regimen (3DR) in ART-Naïve and ART-Experienced Person with HIV (PWH) Across the First Wave of COVID-19 Pandemic"

_viruses, 2024, doi:10.3390/v16121822_

Round 1
Reviewer 1 Report
Comments and Suggestions for Authors
The study by Vergori et al, describe the trends in the use of 2D vs 3D drug regimens of antiretroviral therapy (ART) drugs among ART naïve and ART experienced persons with HIV during the COVID-19 pandemic. This brief report is interesting to note that despite COVID-19 the use of ART of 2D/3D continued among HIV individuals.
The major limitation of this work is that most of the work is descriptive.
This study lacks any conclusive evidence to show any impact of COVID-19 between 2D and 3D regimen individuals.
However, these studies critical to understand and provides a clue for future pandemic scenarios that continuation of 2D/3D ART drugs is essential. The addition of the data on any clinical parameters would be valuable.
Figures 1 and 2 are confusing. Although the legend describes 2D vs 3D the data only shows 2D. The Y-axis should be denoted as 2D vs 3D ART drug regimens.
Author Response
The study by Vergori et al, describe the trends in the use of 2D vs 3D drug regimens of antiretroviral therapy (ART) drugs among ART naïve and ART experienced persons with HIV during the COVID-19 pandemic. This brief report is interesting to note that despite COVID-19 the use of ART of 2D/3D continued among HIV individuals.
The major limitation of this work is that most of the work is descriptive.
This study lacks any conclusive evidence to show any impact of COVID-19 between 2D and 3D regimen individuals.
However, these studies critical to understand and provides a clue for future pandemic scenarios that continuation of 2D/3D ART drugs is essential. The addition of the data on any clinical parameters would be valuable.
Figures 1 and 2 are confusing. Although the legend describes 2D vs 3D the data only shows 2D. The Y-axis should be denoted as 2D vs 3D ART drug regimens.
We appreciate the reviewer’s comment and agree with his/her perspective. The goal of our work was primarily descriptive, specifically to evaluate the frequency of use of available treatment strategies in the ART-naïve and suppressed switch settings during a time of significant uncertainty in healthcare due to the lockdown caused by the COVID-19 pandemic.
In line with your suggestion of adding clinical parameters to the picture, we have now included some information on comorbidity in the descriptive tables.
Regarding the comment on the figures, we agree with the reviewer, and we have now corrected the Y axis to read “Proportion of PWH starting 2DR/ total (%)” for Figure 1 and “Proportion of PWH switching to 2DR /total (%)” for Figure 2.
Reviewer 2 Report
Comments and Suggestions for Authors
Dear authors:
In general, this is a well-written manuscript. Other points in this manuscript needed to be clarified are listed below:
Minor revisions:
1. Materials and methods: Did you exclude experienced and viremic foreign PWH? Because you could have assumed those individuals to be naïve PWH.
2. What is the proportion of PWH in Italy who are positive for HBsAg? I think this factor could a source of bias or limitations in prescribing 2DR or 3DR.
3. The study does not account for all potential confounders, such as socioeconomic status, which may have influenced access to healthcare services during the pandemic and could partially explain the shift in ART prescribing patterns. Maybe you could list this factor as one of limitations.

Author Response
In general, this is a well-written manuscript. Other points in this manuscript needed to be clarified are listed below:
Minor revisions:
- Materials and methods: Did you exclude experienced and viremic foreign PWH? Because you could have assumed those individuals to be naïve PWH.
We believe that it is unlikely that migrants enrolled in Icona had been treated before entering Italy. Icona cohort enrols ART-naive patients as the only inclusion criterion other than HIV-1 infection and the investigators are indeed very careful during the screening, in case of any doubt the PWH should not be enrolled. Nevertheless, being a migrant may be associated with lower access to 2DR and is likely that the pandemic. could have had a larger impact on the migrant population. We have now further adjusted our logistic regression model for nation of birth and results were similar.
In the ART-experienced group, all individuals with detectable viral load >50 copies/mL have been originally excluded. In this revised version, we have further excluded 64 participants for whom their viral load was missing.
- What is the proportion of PWH in Italy who are positive for HBsAg? I think this factor could a source of bias or limitations in prescribing 2DR or 3DR.
We thank the reviewer for pointing this out. Indeed, a HBsAg+ test may preclude the use of some 2DR regimens. In the original submission we had included 2 PWH among the ART-naïve (0.1%) and 262 (2%) HBsAg+ participants in the ART-experienced setting. In this revised version we have now excluded these 264 individuals.
- The study does not account for all potential confounders, such as socioeconomic status, which may have influenced access to healthcare services during the pandemic and could partially explain the shift in ART prescribing patterns. Maybe you could list this factor as one of limitations
We thank the reviewer for this remark which we agree with. Unfortunately, socioeconomics or accurate scores for social deprivations are not collected in the Icona database. However, we do collect nation of birth as well as level of education and employment status which are often used as proxy of socioeconomical status. We have now further controlled for these variables in the logistic regression model and results were almost identical. We have now also added a sentence in the Discussion section to highlight this potential limitation (lines 202-204).
Round 2
Reviewer 1 Report
Comments and Suggestions for Authors
The manuscript is well-written.